# Mortality and Comorbidities in Extremely Low Birth Weight Thai Infants: A Nationwide Data Analysis

**DOI:** 10.3390/children9121825

**Published:** 2022-11-25

**Authors:** Pakaphan Kiatchoosakun, Junya Jirapradittha, Pongsatorn Paopongsawan, Leelawadee Techasatian, Pagakrong Lumbiganon, Kaewjai Thepsuthammarat, Sumitr Sutra

**Affiliations:** 1Department of Pediatrics, Faculty of Medicine, Khon Kaen University, Khon Kaen 40002, Thailand; 2Clinical Epidemiology Unit, Faculty of Medicine, Khon Kaen University, Khon Kaen 40002, Thailand

**Keywords:** extremely low birth weight (ELBW) infants, in-hospital mortality, comorbidities

## Abstract

This is the first nationwide study aimed to evaluate in-hospital mortality and comorbidities of extremely low birth weight (ELBW) infants in Thailand between 2015–2020. Data of ELBW infants were collected from the National Health Coverage Scheme. The incidence of ELBW Thai infants was 1.75 per 1000 live births. Sixty-five percent of ELBW infants were delivered in tertiary-care facilities, with 63% surviving until discharge. In-hospital mortality was 36.9%. Non-invasive respiratory supports were documented in just 17.6% of the study population, whereas total parenteral nutrition was used in 52.3% of neonates. There were several comorbidities, with the three most frequent including respiratory distress syndrome (70.7%), neonatal jaundice (66.7%), and sepsis (60.4%). The median hospitalization cost for one ELBW infant who survived was 296,438.40 baht ($8719). Conclusion: Thailand had an acceptable ELBW infant survival rate (63%), but comorbidities remained particularly severe and cost one hundred times the median hospital cost for one ELBW infant that survived in comparison to a normal newborn infant. Better health outcomes require strategies to raise awareness of the issues and the appropriate implementation of evidence-based solutions, particularly improving neonatal care facilities, as well as early referral of high-risk pregnant women and neonates, which will aid in the future reduction of neonatal morbidities and mortalities.

## 1. Introduction

The neonatal mortality rate is one of the international indicators that reflects the health problems of a country and prematurity remains a significant risk factor for neonatal mortality globally [1,2,3,4,5]. In Thailand, neonatal deaths comprised 58.9% of all deaths in the 1st year of life, of which 41.9% were caused by conditions related to prematurity [1]. Although the subgroup of extremely low birth weight (ELBW) infants (birthweight less than 1000 g) account for a relatively small proportion of all live births, they have a higher mortality rate and more morbidities in survivors, which contribute to their disabilities or long-term problems. The survival rate of ELBW infants in developed countries is dramatically improved [6,7,8,9,10,11,12,13]; however, the survival of this vulnerable population in developing countries remains low due to insufficient resource allocation for appropriate perinatal and neonatal care.

There are limited data reporting mortality and morbidities of ELBW infants from low- and middle-income countries [5,10,14,15,16]. In Thailand, there were reports of this high-risk population from University hospitals [17,18,19] but no detailed information as the overall view at the country level, which is critical to implement programs for improvement of neonatal outcomes.

Since 2002, Thailand’s public health financing schemes have provided universal health care coverage for all Thai populations, and nearly all births (96–99%) took place in hospitals [20]; thus, an evaluation of the health situation among infants from these in-hospital records is the most reliable database available in the country.

This study aimed to evaluate the incidence and in-hospital mortality as well as comorbidities of ELBW infants in Thailand during 2015–2020 from the database of the Bureau of Health Policy and Strategy at the Ministry of Public Health. Factors associated with in-hospital mortality were also investigated.

## 2. Materials and Methods

A descriptive analytic study was conducted on the data of ELBW infants who were born in hospitals in both the public and private sectors in Thailand from January 2015 to December 2020. The data were extracted from the summary discharge of all hospitals under the National Health Coverage Scheme, the Bureau of Health Policy and Strategy at the Ministry of Public Health using the International Statistical Classification of Diseases and Related Health Problems, 10th Revision, Thai Modification (ICD-10-TM). ELBW infants were defined using ICD-10-TM as P070. Data collection included date of birth, date of hospital discharge or death, diagnosis of comorbidities, procedures, length of hospital stay and hospital charges. We calculated the length of hospital stay using the time of birth and the time of first discharge home or death. For infants who were transported to other hospitals after birth, the time spent at other hospitals was included to account for birth hospitalization. Figure 1 demonstrates a flowchart of the present study.

The diagnosis of comorbidity were collected using ICD-10-TM as follows: perinatal asphyxia (P210, P211, P219), respiratory problems including respiratory distress syndrome: RDS (P220), congenital pneumonia (P230-9), pulmonary air leak (P250-8), pulmonary hemorrhage (P260-9), bronchopulmonary dysplasia: BPD (P271), persistent pulmonary hypertension of the newborn: PPHN (P293), neonatal sepsis (P360-9), intraventricular hemorrhage: IVH (P520-9), neonatal hypoglycemia (P704), other transitory disorders of carbohydrate metabolism of newborn approximate synonyms of neonatal hyperglycemia (P708), hyperglycemia R73, disturbance of sodium balance of newborn (P742), hypernatremia E870, hypornatremia E871, neonatal jaundice (P590-9), anemia of prematurity (P612), necrotizing enterocolitis: NEC (P77), patent ductus arteriosus: PDA (Q250), and retinopathy of prematurity: ROP (H351).

The ICD-9 codes for procedures were collected as follows: cardiopulmonary resuscitation (9960), non-invasive respiratory support (9390), continuous invasive mechanical ventilation for less than 96 consecutive hours (9671), continuous invasive mechanical ventilation for 96 consecutive hours or more (9672), intercostal drainage: ICD (3404), packed red cell (PRC) transfusion (9904), parenteral nutrition (9915), intestinal resection (4502-3, 451-4, 454-6), stoma creation (460-6), intestinal anastomosis (459, 4673-9, 4693-4), exploratory laparotomy (450, 5411), percutaneous abdominal drainage (5491), and procedure for PDA closure (3885).

This study was approved by the Khon Kaen University Ethics Committee in Human Research (HE651028) on 22 January 2022.

### Statistical Analysis

Data were described as frequencies and percentages for categorical variables and medians and interquartile ranges (IQRs) for continuous variables. The incidence of ELBW infants per 1000 live births was calculated using the number of total live births from the Department of Provincial Administration, Ministry of Interior, Thailand. Factors associated with mortality, such as sex, diagnosis of comorbidities and procedures, were assessed by Cox proportional hazards regression analysis. All variables were included in the model, and the results were presented as crude and adjusted hazard ratios with 95% confidence intervals (CI) and *p* values. All hypotheses were assessed using two-tailed tests, and *p* values < 0.05 were considered statistically significant. Statistical analyses were carried out using Stata version 10.1 (Stata Corp, College Station, TX, USA).

## 3. Results

During the study period, with a total of 4,014,835 live births, there were 7042 ELBW infants born, giving the overall incidence of ELBW infants at 1.75 per 1000 live births (95% CI 1.71 to 1.79). Among all ELBW infants, 65.1% were born at tertiary-level hospitals, 29.0% at secondary-level hospitals and 2.6% at primary-level hospitals, as shown in Table 1. A total of 2597 ELBW infants died before being discharged. Therefore, the in-hospital mortality rate of ELBW infants was 36.9% (2597/7042), and 83.6% (2170/2597) of all deaths occurred within 28 days of life (neonatal death), of which 56.9% (1478/2597) occurred within 7 days after birth (viz., early neonatal death).

There were many comorbidities found in ELBW infants. The three most common comorbidities were RDS, neonatal jaundice and sepsis, which were diagnosed in 70.7%, 66.7% and 60.4% of the patients, respectively. The other comorbidities included PDA, BPD, anemia of prematurity, congenital pneumonia, IVH (all grades), severe asphyxia and any stages of NEC (Table 2).

Procedures performed in the study population are described in Table 2. Among the respiratory support procedures, mechanical ventilation for more than 96 h was the most frequent procedure performed in the ELBW infants (61.6%). The other common procedures among ELBW infants were PRC transfusion (70%), and total parenteral nutrition (52.3%). Surgical procedures were performed in 8.4% of NEC infants (114/1353) and PDA closure were performed in 12.6% of ELBW infants with PDA (333/2630).

Factors and comorbidities that were found in association with higher probabilities of death were male sex, severe asphyxia, RDS, air leak, pulmonary hemorrhage, PPHN and hyperglycemia; with adjusted hazard ratios (95% CI) and *p*-values, as shown in Table 3. Procedures significantly associated with higher probabilities of death rates were mechanical ventilation, cardiopulmonary resuscitation, and intercostal drainage with adjusted hazard ratios (95% CI) and *p*-values, as shown in Table 3.

The median length of stay of 2597 ELBW infants who died was 6 days (IQR 2–18), and the median hospital charge was 63,249 baht/infant (IQR 27,309–154,787). The median length of stay of 4445 ELBW infants who survived at discharge was 70 days (IQR 47–94), and the median hospital charge was 296,438.40 baht/infant (IQR 168,842–463,709).

## 4. Discussion

This study is Thailand’s first nationwide report on in-hospital mortality and comorbidities among ELBW infants during a 6-year period. Despite the fact that the overall incidence of Thai ELBW infants was 1.75 per 1000 live births, which was quite subtle when compared to the rates in Central America and South Asia, which were 4 and 5 per 1000 live births, respectively [16]. However, when compared to Thailand’s annual dropping birth rate, the frequency of ELBW infants exhibited an increase over the last few years, along with a number of mortalities and comorbidities.

The policy for treating peri-viable infants in Thailand is infants with a birth weight of more than 500 g and a gestational age of more than 24 weeks. The present study discovered 36.9% of in-hospital mortality of ELBW infants’ admission. The majority of deaths (83.6%) occurred within the first 28 days of life (neonatal death), while more than half of non-survivors died within the first 7 days of life.

There were no exclusion criteria in existence. According to the study process flow, the authors began the study design by including all ELBW newborns who were documented in the Thai national database with the ICD-10-TM code P070 (ELBW infant). Despite the fact that some congenital heart disease and chromosomal abnormalities may have an impact on the outcome of an ELBW infant. However, following an extensive analysis, the number of congenital heart diseases and chromosomal abnormalities given in the current study was quite low.

The causes of death within the first 7 days of life in the study population were ELBW extreme preterm (1414 cases), respiratory failure (44 cases), asphyxia (8 cases), sepsis (3 cases), bleeding (2 cases), congenital syphilis (1 case), congenital malformation (4 cases), and others (2 cases).

Several comorbidities occurred in ELBW infants. According to the current study, RDS was the most common respiratory disease, affecting 70.7% of the study population. RDS infants require respiratory support, which can cause lung injury in ELBW infants and lead to the long-term development of BPD. This condition was also identified in 32.4% of the study population, which is comparable to previous studies in other developed countries [8,9] as well as prior Thai studies [17,18,19], which reported a BPD range of 20–42%.

Efforts to minimize lung injury in very preterm infants to prevent BPD by using non-invasive respiratory support such as nasal continuous positive airway pressure (CPAP) have been shown to reduce the need for mechanical ventilation, surfactant therapy, and the incidence of BPD and death. Therefore, the global recommendation has shifted toward non-invasive respiratory support [21]. However, only 17.6% of all infants were successfully supported by non-invasive modes in the present study. As a result, improving non-invasive respiratory support facilities for very preterm infants should be established nationwide, primarily to reduce the incidence of BPD and death.

Other comorbidities reported in the study population included PDA (37.3%), of which 12.6% need surgical ligation, anemia of prematurity (30.7%), and all stages of NEC according to Bell’s classification [22] (19.2%) of which 8.4% required surgical management.

IVH grading was identified according to Papile et al. classification [23]. The present study found that the incidence of IVH (all grades) was 21.2%. This was comparable to the global prevalence of grade 3–4 IVH, which ranged from 5 to 52% [24]. However, IVH in the present study may have been underdiagnosed because routine cranial ultrasonography screening was not available in every hospital. Early detection of IVH may give adequate neonatal care as well as detection of specific neurological abnormalities that require long-term monitoring. As a result, nationwide support facilities for routine cranial ultrasonography screening should be provided, primarily to improve the abilities of IVH early diagnosis.

In Thailand, ROP screening examinations were conducted in accordance with the AAP guideline 2006 [25], which included infants with birth weights of 1500 g or less and gestational ages of 32 weeks or less, for infants with birth weights of 1500 to 2000 g or a gestational age of greater than 32 weeks who have an unstable clinical course. ROP screening will be performed at postnatal age 4–6 weeks. The incidence of ROP in the present study was 16.7%. However, this figure may not accurately reflect the true incidence of ROP in Thailand because some cases had delayed ROP screening. These were caused by a variety of circumstances, including limited number of ophthalmologists, infants with clinically unstable to do an eye examination, and some infants were discharged before ROP screening.

It is recommended to provide total parenteral nutrition in the first few weeks after birth to ensure adequate energy and protein intake in ELBW infants when enteral nutrition is insufficient [26]. However, in the present study, only half of ELBW infants received total parenteral nutrition, and the probability of death in these infants was lower than in those who did not. Despite the fact that there is significant support for early total parenteral nutrition. There were various problems among Thai hospitals in providing early total parenteral nutrition in ELBW neonates, which may have an impact on the outcome of high comorbidities in Thai ELBW infants.

PRC transfusion was a common procedure in infants. According to the current study, 70% of ELBW infants in Thailand received PRC transfusion, which is comparable to 54–90% reported in other studies [27,28]. Transfusion of PRC is essential, particularly in premature infants with insufficient iron reserves. Multiple blood samples taken from very ill infants may result in significant blood loss. Every PRC transfusion attempt, however, may result in complications such as transfusion-related acute lung injury, transfusion-transmitted cytomegalovirus infection, transfusion-associated graft-versus-host disease and hemodynamic instability. Minimizing blood sampling, following standard blood transfusion protocols, and providing enough nutritional and iron support may all help to limit the rate of PRC transfusion. Additionally, performing a delayed cord clamping process during delivery can also reduce the need for PRC transfusion and has excellent empirical support for using this technique among neonates who are at risk of anemia and may require blood transfusion [29,30].

The overall survival rate of Thai ELBW infants was 63.1%. This finding was comparable to previous reports from India and China, which revealed a survival rate of around 62% among ELBW infants [14,31]. In accordance with overall Thai hospitals policy for treating peri-viable infants, which is infants with a birth weight of more than 500 g and a gestational age of more than 24 weeks, Thailand had an acceptable ELBW infant survival rate. However, the comorbidities and burden of complications such as chronic lung disease, cognitive delays, cerebral palsy, and neurosensory deficits remain high [6]. These comorbidities also increased hospitalization costs, and there is still potential for improvement. This issue was also observed in the study population, which found that the median hospitalization cost for one ELBW infant who survived was 296,438.40 baht ($8719), which is approximately 100 times higher and 35 times longer than for a normal-term infant [1]. Therefore, one of the challenges in solving these problems is to recognize and ensure that every pregnant woman and fetus at risk have access to life-saving interventions, including antenatal corticosteroids, antibiotics for preterm, premature rupture of membranes; intrauterine transfer, emergency obstetric management and delivery of ELBW infants at more advanced facilities [32,33].

A meta-analysis also demonstrated that very low birth weight infants born outside a tertiary care hospital are significantly associated with an increased likelihood of neonatal or predischarge death [34]. According to the findings of this study, only 65% of ELBW infants were born in tertiary care settings, which can provide appropriate and advanced care for these high-risk populations, such as resuscitation in the delivery room by trained personnel, stabilization and provision of appropriate respiratory and cardiovascular support, surfactant therapy, and nutritional management. Improving neonatal care facilities, together with early referral of high-risk pregnant woman and neonates, will aid in the reduction of neonatal morbidities in the future. The current national information regarding prognosis after extremely preterm birth allows health care providers and families to plan for potential outcomes and may guide clinical decision-making to optimize care.

This study is the first nationwide report of in-hospital mortality and comorbidities of ELBW infants over a 6-year period. The strengths of this study include the large number of infants and the retrieval of data at the whole country level from the National Health Coverage source available in Thailand. Some limitations of the present study merit consideration. All the information was extracted from ICD-10 and ICD-9 coding by primary through tertiary care hospitals as well as private hospitals throughout the country, so the accuracy of the results depended on diagnosis and coding. Some information was unable to be extracted individually, including gestational age, birth weight, and the mode of delivery. As a result, several critical data were missing. Some misclassification biases might have occurred despite the quality audit process at each hospital level and the external audit by the Bureau of Health Policy and Strategy at the Ministry of Public Health.

## 5. Conclusions

Thailand had an acceptable ELBW infant survival rate (63%), but comorbidities remained particularly severe and cost one hundred times the median hospital cost for one ELBW infant that survived in comparison to a normal newborn infant. Better health outcomes require strategies to raise awareness of the issues and the appropriate implementation of evidence-based solutions, particularly improving neonatal care facilities such as non-invasive respiratory support and total parenteral nutrition in lower-level hospitals, as well as early referral of high-risk pregnant women and neonates, which will aid in the future reduction of neonatal morbidities and mortalities. Further studies in other settings are needed to confirm our findings, and strategies to improve these managements should be emphasized.

## Figures and Tables

**Figure 1 children-09-01825-f001:**
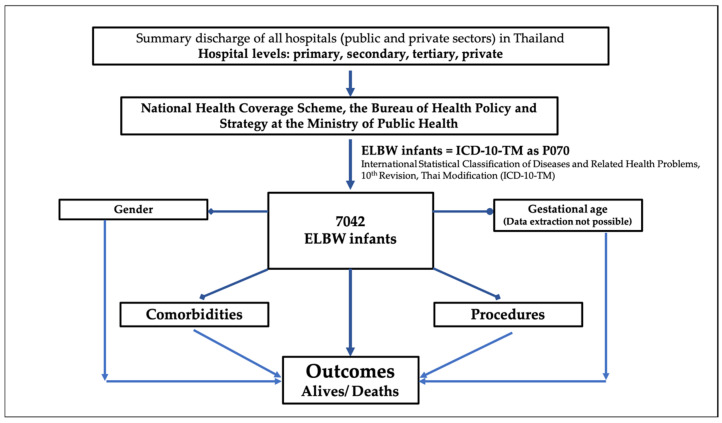
A summary flowchart of the study.

**Table 1 children-09-01825-t001:** Total number of live births and number of extremely low birth weight (ELBW) infants born between 2015 and 2020 classified by year of birth and hospital level.

Year	2015	2016	2017	2018	2019	2020	Total
Total live births	736,352	704,058	702,755	666,109	618,193	587,368	4,014,835
ELBW	1211	1204	1268	1206	1175	978	7042
Per 1000 live births	1.64	1.71	1.80	1.81	1.90	1.67	1.75
Male	586 (48.4%)	597 (49.6%)	639 (50.4%)	605 (50.2%)	622 (52.9%)	492 (50.3%)	3541 (50.3%)
Female	625 (51.6%)	607 (50.4%)	629 (49.6%)	601 (49.8%)	553 (47.1%)	486 (49.7%)	3501 (49.7%)
Hospital level							
Primary	42	31	29	31	25	27	185 (2.6%)
Secondary	332	334	383	479	331	285	2044 (29.0%)
Tertiary	798	798	799	760	792	640	4587 (65.1%)
Private	39	41	57	36	27	26	226 (3.2%)

**Table 2 children-09-01825-t002:** Comorbidities and procedures performed in 7042 ELBW infants.

Comorbidities	N	%
P220 respiratory distress syndrome (RDS)	4982	70.7
P590-9 neonatal jaundice	4694	66.7
P360-9 sepsis	4253	60.4
Q250 patent ductus arteriosus (PDA)	2630	37.3
P271 bronchopulmonary dysplasia (BPD)	2285	32.4
P612 anemia of prematurity	2164	30.7
P230-9 congenital pneumonia	1845	26.2
P520-9 intraventricular hemorrhage (IVH), all grades	1492	21.2
P210 severe asphyxia	1438	20.4
P704 hypoglycemia	1410	20.0
P77 necrotizing enterocolitis (NEC)	1353	19.2
H351 ROP P708, R73 hyperglycemia P250-8 pulmonary interstitial emphysema and air leak	1179 488 695	16.7 6.9 9.9
P260-9 pulmonary hemorrhage	665	9.4
P293 persistent pulmonary hypertension of the newborn (PPHN) P742, E870, E871disturbance of sodium balance of newborn	358 1647	5.1 23.4
Procedures		
**Respiratory support**		
9390 non- invasive respiratory support	1236	17.6
9671 mechanical ventilation <96 h	1467	20.8
9672 mechanical ventilation >96 h	4339	61.6
3891-3 arterial and/or venous catheterization	5440	77.2
9904 blood transfusion	4934	70.0
9915 parenteral nutrition	3686	52.3
9960 cardiopulmonary resuscitation	832	11.8
3404 intercostal drainage **Gastrointestinal tract surgery** 4502-3, 451-6 intestinal resection 460-6 related to stoma creation 459, 4673-9, 4693-4 intestinal anastomosis 450, 5411 exploratory laparotomy 5491 percutaneous abdominal drainage 3885 PDA ligation	468 179 84 98 47 5 63 333	6.6 2.5 1.2 1.4 0.7 0.1 0.9 4.7

**Table 3 children-09-01825-t003:** Factors associated with mortality of ELBW infants by Cox proportional hazards regression analysis.

Variables	Alive N (%)	Death N (%)	Crude HR (95% CI)	Adjusted HR (95% CI)	*p*-Value
Total	4445 (63.1)	2597 (36.9)			
Sex					<0.001
-Female (Ref.)	2364 (67.5)	1137 (32.5)	1	1	
-Male	2081 (58.8)	1460 (41.2)	1.36 (1.26, 1.47)	1.28 (1.19, 1.39)	
Comorbidities					
P210 severe asphyxia					<0.001
-No (Ref.)	3737 (66.7)	1867 (33.3)	1	1	
-Yes	708 (49.2)	730 (50.8)	1.66 (1.52, 1.81)	1.34 (1.22, 1.46)	
P220 RDS					<0.001
-No (Ref.)	1494 (72.5)	566 (27.5)	1	1	
-Yes	2951 (59.2)	2031 (40.8)	1.36 (1.24, 1.49)	1.45 (1.31, 1.59)	
P230-9 congenital pneumonia					0.089
-No (Ref.)	3182 (61.2)	2015 (38.8)	1	1	
-Yes	1263 (68.5)	582 (31.5)	0.63 (0.57, 0.69)	0.92 (0.83, 1.01)	
P250-8 air leak					<0.001
-No (Ref.)	4208 (66.3)	2139 (33.7)	1	1	
-Yes	237 (34.1)	458 (65.9)	2.04 (1.84, 2.26)	1.48 (1.29, 1.70)	
P260-9 pulmonary hemorrhage					<0.001
-No (Ref.)	4193 (65.8)	2184 (34.3)	1	1	
-Yes	252 (37.9)	413 (62.1)	2.04 (1.84, 2.27)	1.95 (1.75, 2.18)	
P271 BPD					<0.001
-No (Ref.)	2373 (49.9)	2384 (50.1)	1	1	
-Yes	2072 (90.7)	213 (9.3)	0.10 (0.08, 0.11)	0.13 (0.11, 0.15)	
P293 PPHN					<0.001
-No (Ref.)	4324 (64.7)	2360 (35.3)	1	1	
-Yes	121 (33.8)	237 (66.2)	1.88 (1.64, 2.15)	1.80 (1.57, 2.07)	
P360-9 sepsis					<0.001
-No (Ref.)	1710 (61.3)	1079 (38.7)	1	1	
-Yes	2735 (64.3)	1518 (35.7)	0.71 (0.66, 0.77)	0.82 (0.75, 0.89)	
P77 NEC					<0.001
-No (Ref.)	3454 (60.7)	2235 (39.3)	1	1	
-Yes	991 (73.2)	362 (26.8)	0.51 (0.46, 0.57)	0.73 (0.65, 0.81)	
Q250 PDA					<0.001
-No (Ref.)	2561 (58.1)	1851 (42.0)	1	1	
-Yes	1884 (71.6)	746 (28.4)	0.49 (0.45, 0.53)	0.72 (0.65, 0.78)	
P708, R73 hyperglycemia					<0.001
-No (Ref.)	4207 (64.2)	2347 (35.8)	1	1	
-Yes	238 (48.8)	250 (51.2)	1.53 (1.35, 1.75)	1.42 (1.24, 1.62)	
P742, E870, E871 disturbance of sodium balance					0.079
-No (Ref.)	3325 (61.6)	2070 (38.4)	1	1	
-Yes	1120 (68.0)	527 (32.0)	0.66 (0.60, 0.72)	0.91 (0.83, 1.01)	
Procedures					
9390 non-invasive respiratory support (Ref.)	989 (80.0)	247 (20.0)	1	1	<0.001
9671 mechanical ventilation < 96 h	475 (32.4)	992 (67.6)	5.04 (4.39, 5.80)	3.71 (3.20, 4.30)	
9672 mechanical ventilation > 96 h	2981 (68.7)	1358 (31.3)	1.02 (0.89, 1.17)	1.35 (1.16, 1.57)	
9960 CPR					<0.001
-No (Ref.)	4257 (68.6)	1953 (31.5)	1	1	
-Yes	188 (22.6)	644 (77.4)	3.39 (3.10, 3.70)	2.18 (1.98, 2.40)	
3404 ICD					0.008
-No (Ref.)	4289 (65.2)	2285 (34.8)	1	1	
-Yes	156 (33.3)	312 (66.7)	2.01 (1.79, 2.27)	1.24 (1.06, 1.45)	
9904 PRC transfusion					<0.001
-No (Ref.)	1153 (54.7)	955 (45.3)	1	1	
-Yes	3292 (66.7)	1642 (33.3)	0.42 (0.39, 0.46)	0.62 (0.56, 0.67)	
9915 parenteral nutrition					<0.001
-No (Ref.)	1883 (56.1)	1473 (43.9)	1	1	
-Yes	2562 (69.5)	1124 (30.5)	0.49 (0.46, 0.53)	0.69 (0.64, 0.75)	
Gastrointestinal tract surgery					0.008
-No (Ref.)	4349 (63.4)	2514 (36.6)	1	1	
-Yes	96 (53.6)	83 (46.4)	0.92 (0.74, 1.15)	1.37 (1.08, 1.72)	
3885 PDA ligation					0.002
-No (Ref.)	4171 (62.2)	2538 (37.8)	1	1	
-Yes	274 (82.3)	59 (17.7)	0.31 (0.24, 0.40)	0.64 (0.49, 0.84)	

N: number: CI: confidence interval; HR: hazard ratio; Ref: reference; RDS: respiratory distress syndrome; BPD: bronchopulmonary dysplasia; PPHN: persistent pulmonary hypertension of the newborn; NEC: necrotizing enterocolitis; PDA: patent ductus arteriosus; CPR: cardiopulmonary resuscitation; ICD: intercostal drainage; PRC: packed red cell.

## Data Availability

The datasets used and analyzed during the current study are from Thailand National Health Coverage but restrictions apply to the availability of these data, which were used under license for the current study, so are not publicly available. Data are however available from the authors upon reasonable request and with permission of Thailand National Health Coverage.

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
