# Peer review of "Mortality and Comorbidities in Extremely Low Birth Weight Thai Infants: A Nationwide Data Analysis"

_children, 2022, doi:10.3390/children9121825_

Round 1

Reviewer 1 Report

The survival rate of extremely low birth weight (ELBW) infants in developed countries improved, however, in the developing world remains low due to insufficient resource allocation for perinatal and neonatal care.

Pakaphan Kiatchoosakun et al in this paper entitled “Mortality and comorbidities in extremely low birth weight Thai infants: A nationwide data analysis” aimed to evaluate in-hospital mortality and comorbidities of ELBW infants in Thailand.

Data obtained from ELBW infants (born between 2015-2020) were collected from the summary discharge of all hospitals under the National Health Coverage Scheme. In-hospital mortality, comorbidities, length of hospital stay, and cost of hospital charges were analyzed.

The incidence of ELBW Thai infants was 1.75 per 1,000 live births (7,042 ELBW among 4,014,835 live births). Sixty-five percent of ELBW infants were delivered in tertiary-care centers, and 63% survived until discharge. The diagnosis of comorbidity was collected using ICD-10-TM. Factors associated with mortality, such as sex, diagnosis of comorbidities, and procedures, were assessed by Cox proportional hazards regression analysis. There were several comorbidities, the three most frequent including respiratory distress syndrome (70.7%), neonatal jaundice (66.7%), and sepsis (60.4%). Mechanical ventilation for more than 96 hours was the most frequent procedure performed in ELBW infants (61.6%). The other common procedures among ELBW infants were PRC transfusion (70%), and parenteral nutrition (52.3%). Non-invasive respiratory supports were documented in 17.6% of the study population. The median length of hospitalization who survived at discharge was 70 days, and the cost for one ELBW infant who survived was $ 8,719 in Thailand, and nearly all births (96–99%) took place in hospitals. The mortality rate of ELBW infants was 36.9%, and the majority of deaths (83.6%) occurred within the first 4 weeks of life.

The present paper provides important data which are characteristic of Thailand and useful to compare the health system capabilities to other developing countries.

However, there are several limitations, and the following questions need to be answered.

1. The gestational age was not taken into consideration.

2. The mean birth weight and the weight distribution are also very important when the survival rate, the neonatal morbidities are discussed.

3. What is the policy for treating pereviable infants? Is there any limit to starting active treatment?

4. ROP is listed as an abbreviation, but nothing was mentioned about the disease itself, is there any screening policy during hospitalization?

5. Hypoglycemia is common among ELBW infants, it was analyzed, but hyperglycemia occurs in at least 30% of these infants, and the incidence of dysnatremia is about 70%. None of them were investigated although these disturbances may play an important role during postnatal development, for example, risk factors for BPD, ROP, IVH.

The above-mentioned limitations are influencing not just the survival rate, but the quality of life also. I suggest collecting the missing data and resubmitting this article again.

Author Response

Dear editors,

            Thank you for spending the time to read our manuscript. The following are point-by-point responses to reviewer comments:

Reviewer#1

The present paper provides important data which are characteristic of Thailand and useful to compare the health system capabilities to other developing countries.

However, there are several limitations, and the following questions need to be answered.

  1. The gestational age was not taken into consideration.

            Response:

            The authors agree with the reviewer's recommendation. However, it was a limitation of our study design that began by including all ELBW newborns who were documented in the Thai national database with the ICD-10-TM code P070 (ELBW infant).

            The Thai National Health Coverage Scheme's data source system, the Bureau of Health Policy and Strategy of the Ministry of Public Health, was unable to extract individual information that was not recorded in ICD pattern. As a result, gestational age and birth weight, were missing. We completely agree on the importance of this information, hence we have included this limitation in the revised manuscript to allow for further research into this value point.

  1. The mean birth weight and the weight distribution are also very important when the survival rate, the neonatal morbidities are discussed.

            Response:

            The birth weight of the individual could not be extracted while start searching with the ICD-10-TM code P070 (ELBW newborn). We are totally in agreement with the importance of this information, so we have added this limitation in the revised manuscript to allow for additional investigation into this value point.

  1. What is the policy for treating periviable infants? Is there any limit to starting active treatment?

            Response:

            The policy for treating peri-viable infants in Thailand is infants with a birth weight of more than 500 g and a gestational age of more than 24 weeks. We completely agree on the importance of this information, hence we have included this information in the revised manuscript.

  1. ROP is listed as an abbreviation, but nothing was mentioned about the disease itself, is there any screening policy during hospitalization?

            Response:

            We apologized for the lack of this information. In Thailand, ROP screening examinations were conducted in accordance with the AAP guideline 2006, which included infants with birth weights of 1,500 g or less and gestational ages of 32 weeks or less, for infants with birth weights of 1,500 to 2,000 g or a gestational age of greater than 32 weeks who have an unstable clinical course. ROP screening will be performed at postnatal age 4-6 weeks. However, there were some cases of delayed ROP screening caused by a variety of factors, including an infant who was clinically unstable and unable to perform an eye examination, an ophthalmologist who was unavailable in some hospitals, and some infants who were discharged before the time of ROP screening.

            We completely agree on the importance of this information, hence we have included this information in the revised manuscript.

  1. Hypoglycemia is common among ELBW infants, it was analyzed, but hyperglycemia occurs in at least 30% of these infants, and the incidence of dysnatremia is about 70%. None of them were investigated although these disturbances may play an important role during postnatal development, for example, risk factors for BPD, ROP, IVH.

            Response:

            The authors took a step back and investigated hyperglycemia and dysnatremia in the study population. The current study found 6.9% and 23.4% hyperglycemia and disturbance of sodium balance in newborn, respectively. The low percentage of these comorbidities can be explained by the limitations of coding accuracy due to unrecognized of the problems, which is almost certainly lower than the actual situation. This information was also included in Table 2 and 3 of the revised manuscript. Thank you for your suggestion.

We hope that our revised manuscript will meet the standards for publication in Children to provide further information on the Thai national database of ELBW infants.

Sincerely yours,

Associate Professor Pakaphan Kiatchoosakun

(Corresponding author)

Reviewer 2 Report

The authors investigated the nationwide data in Thailand to clarify the mortality and comorbidities among ELBW infants for six years. They found interesting results stating the mortality, the incidence of severe comorbidities (pneumonia, NEC, sepsis, and so on), and the method of primary care (respiratory support, parenteral nutrition) of admission ELBW infants. However, I think some crucial information is not available.

 Major points

ü  I think it will be easier to understand your study with a flowchart. What were the exclusion criteria of this study? Did you exclude infants with congenital heart diseases and chromosomal abnormalities? Are there any ELBW infants with missing ICD data?

ü  I think gestational age, body weight at birth, small for gestational age status, mode of delivery, and inborn/outborn birth status are the essential factors for mortality and comorbidities in ELBW infants. If possible, please add these data.

ü  The diagnostic criteria and grade (severity) of the comorbidities are unclear. Do you have any data for surgical procedures, especially in PDA and NEC?

ü  What are the criteria for “parenteral nutrition”? Did you include infants with only glucose infusion?

ü  Please add information on the course of death within the first 7 days of life.

Author Response

Dear editors,

            Thank you for spending the time to read our manuscript. The following are point-by-point responses to reviewer comments:

Reviewer#2

  1. I think it will be easier to understand your study with a flowchart. What were the exclusion criteria of this study? Did you exclude infants with congenital heart diseases and chromosomal abnormalities? Are there any ELBW infants with missing ICD data?

Response:

  • I think it will be easier to understand your study with a flowchart.
    • Response:

The authors agree with the reviewer's recommendation. The study flow chart has now been developed and submitted alongside the revised manuscript.

  • What were the exclusion criteria of this study? Did you exclude infants with congenital heart diseases and chromosomal abnormalities?
    • Response:

      There were no exclusion criteria in existence. According to the study process flow, we began the study design by including all ELBW newborns who were documented in the Thai national database with the ICD-10-TM code P070 (ELBW infant).

      The authors agree with the reviewer's suggestion that congenital heart disease and chromosomal abnormalities may have an impact on the outcome of an ELBW infant. However, following an extensive analysis, as suggested by the reviewers, the number of congenital heart diseases and chromosomal abnormalities given in the current study was quite low, (4 in 1,414 of early neonatal deaths cases). This information has been included in the revised manuscript. Thank you very much for your valuable advice.

  • Are there any ELBW infants with missing ICD data?
    • Response:
    • There was very little chance of missing ICD data for ELBW newborns because it is a critical ICD-10 code to capture in our country.
    • All the information was extracted from ICD-10 and ICD-9 coding by primary through tertiary care hospitals as well as private hospitals throughout the country and 96-99%of births took place in hospitals, so the accuracy of the results depended on diagnosis and coding.

  1. I think gestational age, body weight at birth, small for gestational age status, mode of delivery, and inborn/outborn birth status are the essential factors for mortality and comorbidities in ELBW infants. If possible, please add these data.

            Response:

            The authors agree with the reviewer's recommendation. However, it was a limitation of our study design that began by including all ELBW newborns who were documented in the Thai national database with the ICD-10-TM code P070 (ELBW infant).

            The Thai National Health Coverage Scheme's data source system, the Bureau of Health Policy and Strategy of the Ministry of Public Health, was unable to extract individual information that was not recorded in ICD pattern. As a result, several critical data, as highlighted by the reviewer, were missing. We completely agree on the importance of this information, hence we have included this limitation in the revised manuscript to allow for further research into this value point.

  1. The diagnostic criteria and grade (severity) of the comorbidities are unclear. Do you have any data for surgical procedures, especially in PDA and NEC?

            Response:

            Thank you for bringing up such an important point. In the revised manuscript, we added comorbidity criteria such as Modified Bell staging criteria for necrotizing enterocolitis (NEC) in newborns and IVH grading, as well as updated the relevant references. Surgical procedures were performed in 8.4% of infants with NEC (114/1,353), while PDA ligation was undertaken in 12.6% of ELBW infants with PDA (333/2,630). This information was also included in the revised manuscript. Thank you for your recommendation.

  1. What are the criteria for “parenteral nutrition”? Did you include infants with only glucose infusion?

            Response:

            In the current study, those who received parenteral nutrition received additional nutritional support and calories, including amino acids, lipids, macro and micronutrients in addition to glucose infusion alone. As a consequence of the reviewer's recommendation, we provided this clarification in the revised manuscript and used the term "total parenteral nutrition" instead of "parenteral nutrition" to prevent misunderstandings. Thank you for your value recommendation.

  1. Please add information on the course of death within the first 7 days of life.

            Response:

            The authors agree with the reviewer's recommendation. The course of death within the first 7 days of life were ELBW extreme preterm (1414 cases), respiratory failure (44 cases), asphyxia (8 cases), sepsis (3 cases), bleeding (2 cases), congenital syphilis (1 case), congenital malformation (4 cases), and others (2 cases). This information was also included in the revised manuscript. Thank you for your recommendation.

            We hope that our revised manuscript will meet the standards for publication in Children to provide further information on the Thai national database of ELBW infants.

Sincerely yours,

Associate Professor Pakaphan Kiatchoosakun

(Corresponding author)

Round 2

Reviewer 1 Report

The authors made the suggested changes in their manuscript, answered my questions, and the reviewer accepts the provided answers. However, the limitations are still there but there is no way to recover certain data from the database.

Reviewer 2 Report

Thank you for the revision according to my comments.